# YOLO-Based UAV Technology: A Review of the Research and Its Applications

Chunling Chen [1], Ziyue Zheng [1], Tongyu Xu [1], Shuang Guo [1], Shuai Feng [1], Weixiang Yao [1,*] and Yubin Lan [2,*]

[1] College of Information and Electrical Engineering, Shenyang Agricultural University, Shenyang 110866, China
[2] College of Electronic Engineering and Artificial Intelligence, South China Agricultural University, Guangzhou 510642, China
* Correspondence: yaoweixiang@syau.edu.cn (W.Y.); ylan@scau.edu.cn (Y.L.)

**Abstract:** In recent decades, scientific and technological developments have continued to increase in speed, with researchers focusing not only on the innovation of single technologies but also on the cross-fertilization of multidisciplinary technologies. Unmanned aerial vehicle (UAV) technology has seen great progress in many aspects, such as geometric structure, flight characteristics, and navigation control. The You Only Look Once (YOLO) algorithm was developed and has been refined over the years to provide satisfactory performance for the real-time detection and classification of multiple targets. In the context of technology cross-fusion becoming a new focus, researchers have proposed YOLO-based UAV technology (YBUT) by integrating the above two technologies. This proposed integration succeeds in strengthening the application of emerging technologies and expanding the idea of the development of YOLO algorithms and drone technology. Therefore, this paper presents the development history of YBUT with reviews of the practical applications of YBUT in engineering, transportation, agriculture, automation, and other fields. The aim is to help new users to quickly understand YBUT and to help researchers, consumers, and stakeholders to quickly understand the research progress of the technology. The future of YBUT is also discussed to help explore the application of this technology in new areas.

**Keywords:** YOLO; UAV; object detection; interdisciplinary; application

## 1. Introduction

As science and technology develop, new ways of living and working emerge, and new technologies with higher value gradually replace old ones. The value of new technologies is not only the innovation and development of the technology but whether the technology is effective in improving productivity and making a contribution to human society, i.e., the application of new technologies to change traditional ways of addressing existing problems to improve social productivity. Today, interdisciplinary or multifield cooperation is a trendy topic; that is, mature technologies in multiple fields are combined to become a new technology, and the advantages of various technologies are retained to compensate for the disadvantages. The integration of existing technologies in multiple fields can not only quickly generate new methods and ideas to address existing problems but can also greatly reduce resource consumption. Currently, unmanned aerial vehicles (UAVs) or aerial robotics are in a period of rapid development [1], and target detection performance based on the You Only Look Once (YOLO) algorithm [2] has reached a high level in industry. The algorithm still needs to be modified and improved [3]. UAVs can carry a variety of devices to accomplish different tasks. Examples of these tasks include spraying liquid medicine [4], mapping [5], logistics transportation [6], disaster management [7], aerial photography [8], and sowing fertilizer or seeds [9]. Object detection technology based on the YOLO algorithm has been able to achieve human behavior analysis [10], face mask recognition [11], medical diagnosis analysis [12], autonomous driving [13], traffic

assessment [14], multitarget tracking [15], and robot vision [16]. However, UAVs face complex scenarios or work with the need to maintain good data communication with ground control terminals, so the innovation and development of UAV technology may be limited by certain application environments. However, UAVs face complex scenarios or work with the need to maintain good data communication with ground control terminals, so the innovation and development of UAV technology may be limited by certain application environments. Moreover, object detection technology based on the YOLO algorithm needs to be deployed into high-performance processors and be used in conjunction with image or video data, which places certain requirements on the scenarios where it is used. These two technologies can be combined to create a new technology—YOLO-based UAV technology (YBUT). UAVs provide more application scenarios for the YOLO algorithm, and the YOLO algorithm can assist UAVs in completing more novel tasks. In this way, drone technology and the YOLO algorithm can further facilitate people's daily lives while contributing to the productivity of their respective industries.

A UAV is often defined as an unmanned flying device that can either fly autonomously according to a course or program established within the system or can be manually controlled by the controller. UAVs can be classified into various types depending on various parameters. In recent years, as a hot spot in the new round of global scientific and technological revolution and industrial revolution, UAVs have been able to replace most of the tasks that used to be completed by manned aircraft. At the same time, as UAV technology continues to mature, the number of UAVs in countries around the world is increasing every year, and according to Global commercial drone annual sales and sales statistics [17], as illustrated in Figure 1, there will be approximately 2,679,000 UAVs in the world by 2025, with a market size worth approximately USD 5.3 billion. With such a large number of UAVs worldwide, it may be possible to make UAVs more valuable if they can be used as aerial platforms to deploy YOLO algorithms.

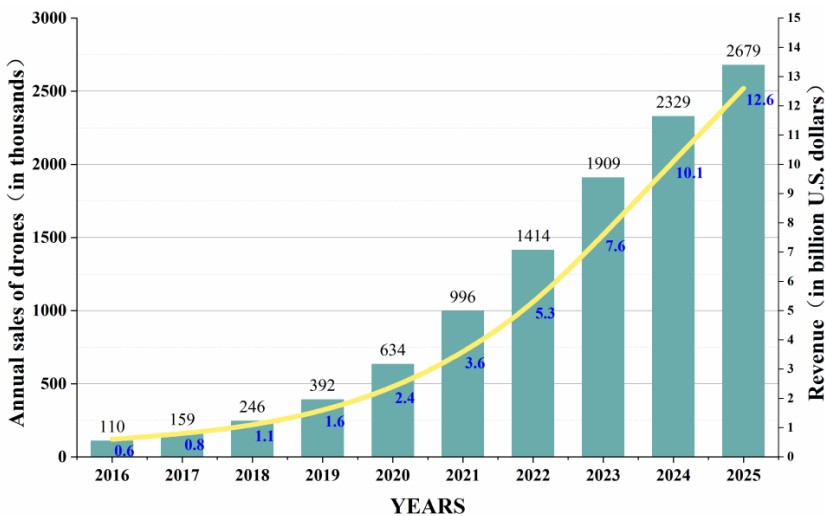

**Figure 1.** Global commercial drone annual sales and sales statistics [17].

YOLO is a widely used deep learning algorithm, because it is a classification-/regression-based object detection method, giving the algorithm its core strengths: a very simple structure, small model size, and fast computational speed. After seven years of development since the introduction of YOLO (as of February 2023), researchers released seven versions of the YOLO algorithm [18–22]. After the YOLO algorithm was popularized, researchers and users improved it for various applications due to its openness and ease of secondary development, and various revisions were introduced, such as YOLODrone [23], YOLOv4_Drone [24], VIT-YOLO [25], YOLO-RTUAV [26], YOLO-Neck [27], and YOLOv7-DeepSORT [28]. The mechanism of the YOLO-based object detection algorithm is that the input image is resized to the same size, and then the image is divided into a total of S × S network cells of equal

size, and each individual network cell can detect objects within it. If the center of a detected target falls into a network cell, that network cell will make a prediction about the target. Each network cell may have N detection boxes, each of which not only calculates its own position but also makes a prediction score. The score represents the likelihood that a detection target is present in the predicted network cell. As there may be multiple boxes in a network cell, YOLO will automatically select the highest-scoring target category for prediction (see Figure 2).

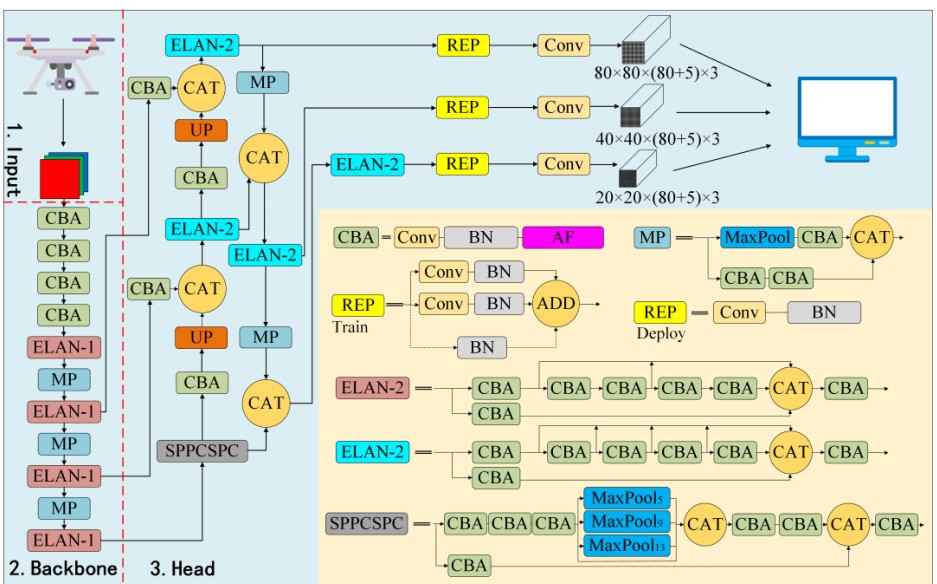

**Figure 2.** YOLOv7-based UAV technology architecture diagram (BN: batch normalization layer; AF: activation function layer).

When UAVs are used for operations in various industries and when an operation involves target identification and positioning, manual operation is often required to confirm the goal location and operation location through real-time images transmitted by UAV cameras, such as transport UAVs for precise parcel delivery. The process is tedious and cumbersome and requires a high level of UAV handling skills, which can easily lead to accidents and financial losses. Although YOLO-based target detection technology emerged late, it has been continuously improved and developed by researchers and already has a high accuracy rate in the fields of object detection and image recognition; the detection speed and accuracy have reached the forefront of the industry while playing an important role in the application of target recognition in UAV aerial images [29]. Khang et al. [30] conducted experiments on the VisDrone2019 dataset containing 96 videos and 39,988 annotated frames and evaluated deep learning detectors with FPS and mAP as evaluation metrics, including Faster R-CNN, RFCN, SNIPER, YOLO, RetinaNet, and CenterNet. Ammar et al. [31] evaluated the performance of convolutional neural network models, such as Faster R-CNN, YOLOv3, YOLOv4, and EfficientDet, using IoU, precision, recall, F1, AP, and mAP as evaluation metrics. The experimental results showed that YOLO is ideal for real-time target detection applications. If the UAV is equipped with an embedded processor deploying the YOLO algorithm, object detection recognition can be realized on real-time footage from the UAV camera, turning the two steps of UAV acquisition and computer detection into simultaneous UAV acquisition and detection, which greatly saves operational time and improves operational efficiency. The improvement in the level of autonomous target recognition by drones can strongly promote the automation or unmanned operation of drones in most industries.

Over the past few years, YBUT has become a popular research area of interest, but the application scenarios and impact of the technology have yet to be enhanced; a summary overview of the recent state of the application in this technology area is lacking. This

paper, therefore, presents the history of YBUT and provides an overview of case studies of the application of YBUT in several industry sectors, intending to provide researchers, beginners, and consumers with a better understanding of the field, as well as a reference for the research and application of the technology in new areas. Here, we also discuss the direction of the technology and provide an outlook on its application.

## 2. Survey Methodology

In this section, we explain the methodology and the idea behind the selection of the papers studied and the main areas of application of YBUT. To screen the literature efficiently and quickly for papers within the scope of this overview, a clear and simple screening process was identified for the published literature, and its methodology is explained, together with an analysis of the main research directions of interest in international and Chinese journals.

### 2.1. Screening Methods for Research Papers Related to YBUT

To search for high-impact research/papers on aerial robots or UAVs that use deep learning YOLO models/algorithms, many of the keywords come from top journals and conferences, including the Web of Science Core Collection, KCI (Korean Journal Database), MEDLINE®, SciELO Citation Index, and China National Knowledge Infrastructure indexed journals, among others. The collected keywords were grouped into A1, A2, A3, and A4 groups and searched in various search engines; then, the results were then filtered for the next step. The keyword groupings used and the detailed search method for the articles are shown in Figure 3.

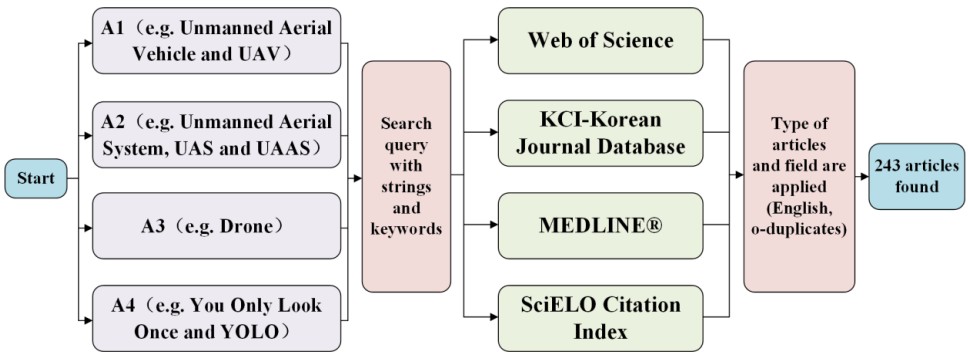

**Figure 3.** Literature search methods.

With the Web of Science search engine as an example, after searching for the above keywords, 243 articles in 41 research fields were collected as of February 2023, containing the types of papers, conferences, revisions, letters, etc., with the specific subject directions shown in Table 1.

Based on the applicability of the filtered articles, we did not consider multiple topics, such as revisions and letters. The articles were also rigorously reviewed for content to remove articles that did not contribute to the topic of this research review, with a focus on checking the image data within the article and the dataset used. At the same time, the articles were verified and analyzed for algorithmic improvements and innovations, and some articles were selected that were progressive or identified as implementable for the development of the relevant industry. Finally, the introduction, discussion, summary, and outlook of the articles, after the screening process was completed, were checked and categorized. Each step of the methodology used in the screening process of the required articles is shown in Figure 4.

**Table 1.** Specific subject directions of the screened articles.

| No. | Research Fields | No. | Research Fields | No. | Research Fields | No. | Research Fields |
|---|---|---|---|---|---|---|---|
| 1 | Engineering | 12 | Transportation | 23 | Plant Sciences | 34 | Architecture |
| 2 | Computer Science | 13 | Optics | 24 | Forestry | 35 | Behavioral Sciences |
| 3 | Automation Control Systems | 14 | Physical Sciences Other Topics | 25 | Physical Geography | 36 | Biodiversity Conservation |
| 4 | Communication | 15 | Geology | 26 | Spectroscopy | 37 | Geography |
| 5 | Instruments Instrumentation | 16 | Environmental Sciences Ecology | 27 | Geochemistry Geophysics | 38 | Neurosciences Neurology |
| 6 | Robotics | 17 | Energy Fuels | 28 | Materials Science | 39 | Parasitology |
| 7 | Business Economics | 18 | Construction Building Technology | 29 | Operations Research Management Science | 40 | Radiology Nuclear Medicine Medical Imaging |
| 8 | Mathematics | 19 | Agriculture | 30 | Zoology | 41 | Mechanics |
| 9 | Imaging Science Photographic Technology | 20 | Mathematical Computational Biology | 31 | Science Technology Other Topics | | |
| 10 | Telecommunications | 21 | Physics | 32 | Remote Sensing | | |
| 11 | Chemistry | 22 | Acoustics | 33 | Fisheries | | |

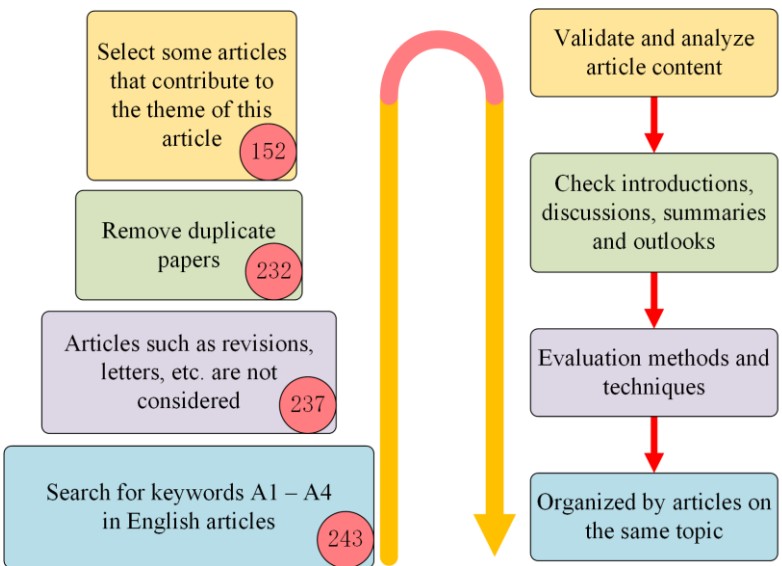

**Figure 4.** Literature survey methodology.

### 2.2. Research Topics Utilizing YBUT

Through the method described above, the search results were analyzed by using both English and Chinese search engines, such as Web of Science and China National Knowledge Infrastructure, to obtain the main research themes of English and Chinese journals in the relevant fields. Computer vision technology has developed a great variety of algorithms to date, among which the YOLO algorithm was proposed in 2016 and then first applied in 2017 by Jiang et al. [32], who combined the YOLO algorithm with UAVs. Since then, the YOLO algorithm and UAV fusion technology have been continuously developed, and there has been a surge in related research results or applications. The technology has also moved from an exploratory experiment to an academic research hotspot (see Figure 5).

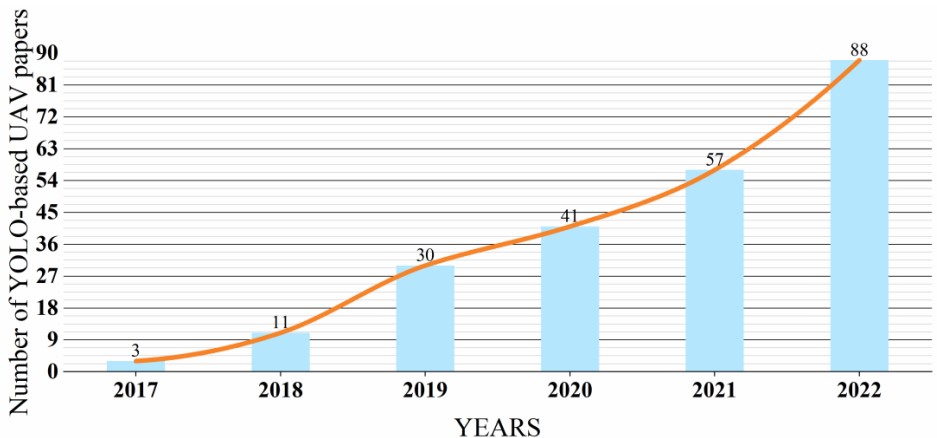

**Figure 5.** Number of papers published in top journals and conferences (2017–2022).

Based on our survey of YBUT application areas, the information for popular topics in this field in English journals is summarized in a pie chart, as shown in the survey results in Figure 6. As seen from the pie chart, the popular topics are mainly in the industries of technical studies, engineering, and transportation, and the number of published papers or conference literature represents the interest of researchers. We also surveyed Chinese journals on popular topics in this area and found that they focus more on the technical studies, engineering, and automation sectors. As UAV technology and YOLO algorithms continue to evolve, this technology is beginning to be explored in most areas, and in a few areas, there have been some successes. The development and research of YBUT have been hot topics in top journals and conferences, and now the practical application of the technology is gradually attracting their interest.

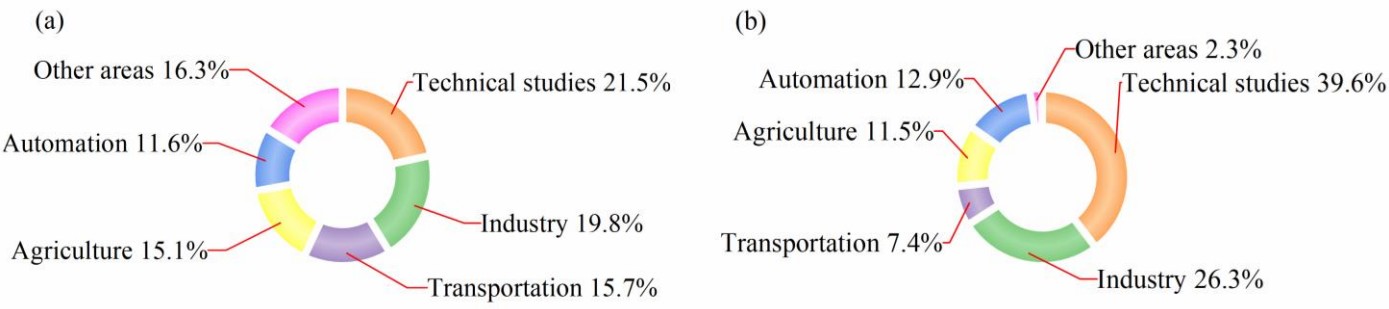

**Figure 6.** Survey of application areas for YBUT: (**a**) Popular areas of interest in English journals, (**b**) Popular areas of interest for Chinese journals.

## 3. YBUT Development

YBUT advances without the support of high-performance computer processors. UAVs have moved from being operated manually by remote control to now being controlled automatically by computers, and image recognition has moved from being run by computer systems to now being run by onboard embedded systems for real-time detection and recognition. Each of these technological advances has taken the application of technology to a new level in some areas.

### 3.1. Early Development of YBUT

At the beginning of the research on YBUT, the technology was proposed because of a technological fusion between UAV technology and YOLO algorithms in the context of a trend towards cross-disciplinary development. Among other things, UAV technology research began in the 1920s and has been developed, to date, with successful applications in agriculture, surveillance, monitoring, traffic construction, system transportation, system

inspection, etc. The YOLO algorithm was proposed in 2016 [2], and after several improvements, it has reached the forefront of the object recognition field in terms of detection speed, detection accuracy, and recognition classification.

The application of YBUT in real production operations started in 2017. In the early stages of the YBUT application, the main working mechanism was image or video data acquisition by UAVs, followed by object detection, identification, and classification by computers running YOLO-based object detection algorithms. To explore methods to detect vehicles from UAV-captured images for application in traffic monitoring and management, and as deep learning algorithms have shown significant advantages in target detection, researchers have tried to apply YOLO-based object detection algorithms to vehicle detection in UAV images. Jiang et al. [32] constructed a multisource data acquisition system by integrating a thermal infrared imaging sensor and a visible-light imaging sensor on a UAV, corrected and aligned the images through feature point extraction and single response matrix methods, and then performed image fusion on the multisource data. Finally, they utilized a deep learning YOLO algorithm for data training and vehicle detection (see Figure 7). The experimental results found that the inclusion of a thermal infrared image dataset could improve the accuracy of vehicle detection and verified that the YOLO framework is an advanced and effective framework for real-time target detection. The first attempt to combine and apply the YOLO algorithm with UAV technology by Jiang et al. [32] demonstrated the usability of YOLO-based UAV technology. Although the detection performance of the early YOLOv1 algorithm was not very good, the experimental results were relatively satisfactory as the first exploration of the technology and the innovative incorporation of thermal infrared image data.

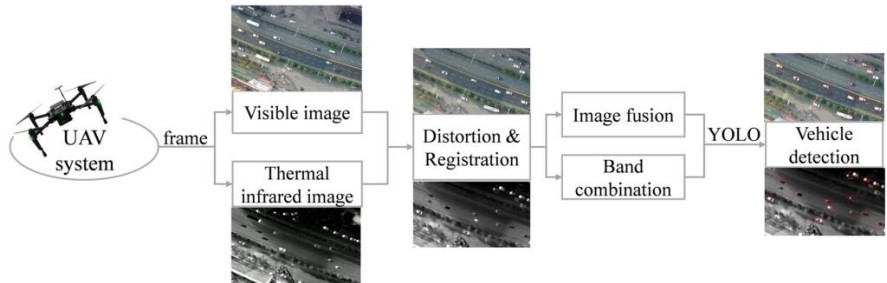

**Figure 7.** Flowchart of the proposed method by Jiang et al. [32].

Based on this study, Xu et al. [33] proposed an improved algorithm for small vehicle detection based on YOLOv2, whose detection structure model is shown in Figure 8. Compared with the YOLOv2 model structure, the algorithm adds an additional feature layer that can reach 1/32 of the input image in size, making the algorithm more adept at detecting small targets and having higher localization accuracy than YOLOv2. This research has greatly contributed to researchers' understanding of YBUT and has also inspired researchers to make targeted improvements to the YOLO algorithm structure when carrying out applications in this field. Since then, Ruan et al. [34] and Yang et al. [35] have further explored the application of YBUT in other fields.

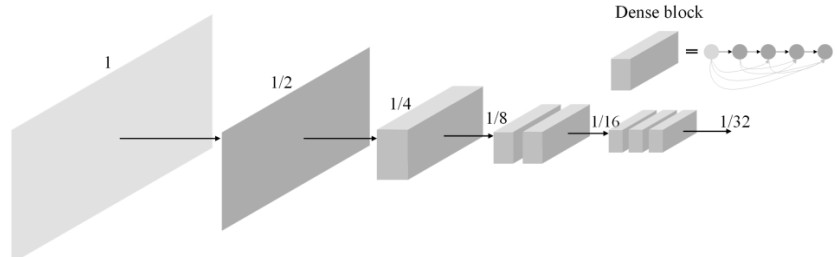

**Figure 8.** Dense YOLO network model [33].

In addition, Ruan et al. [34] attempted to use a deep learning and vision-based drogue detection and localization method to address the accurate detection and localization of fog droplets for autonomous aerial refueling of UAVs in complex environments. They used the trained YOLO algorithm for cone trace detection, the least squares ellipse fit to determine the long semiaxis of the ellipse after determining the fiducial location, and, finally, a monocular vision camera for vertebral drogue localization (see Figure 9). The simulation experimental results show that the method can not only correctly identify cones in complex environments but also accurately locate cones in a range of 2.5–45 m, indicating that the YOLO method has good results for target object detection and localization in various complex environments. Yang et al. [35] investigated a method to achieve real-time pedestrian detection and tracking on a mobile platform with multiple disturbances; they attempted to use a UAV hovering in the air for data acquisition of special targets while using a ground station deployed with YOLOv2 to accept video streams from the UAV for analysis and detection. The results of outdoor pedestrian detection experiments showed the robustness of the method when the brightness varied and pedestrians continued to interfere, demonstrating that this is a stable method for exceptional pedestrian tracking on UAV platforms. Most of these early studies explored simple applications of the fusion of the two technologies due to the lack of maturity of the technology fusion application, but the information gained from the research is of greater reference value for subsequent research on YBUT. An increasing number of researchers are focusing on and exploring the field of YOLO-based UAVs, continuing to drive progress in the development of the field, and a new generation of YBUT has emerged as the performance of high-performance computer processors increases while the size of the hardware decreases.

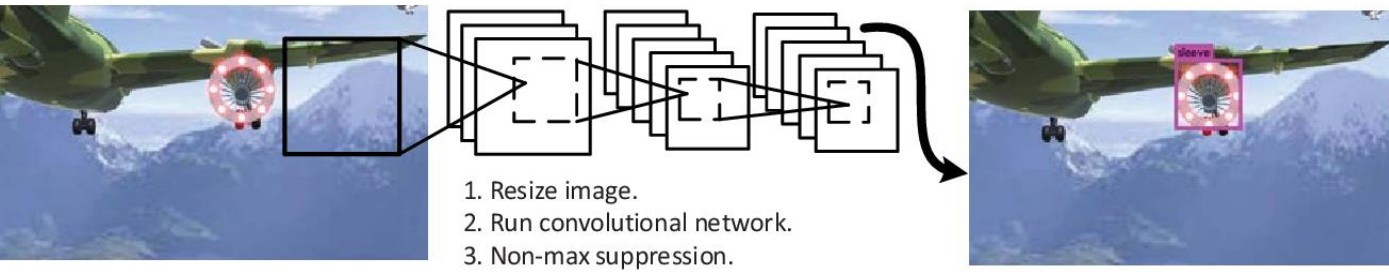

**Figure 9.** Drogue detection method [34].

### 3.2. YBUT Develops by Leaps and Bounds

As YBUT continues to evolve, a new generation has emerged in which the UAV is equipped with a high-performance processor rich in computing resources, within which YOLO-based object detection algorithms are deployed, allowing the processor to detect, to identify, and to classify mission objects in real time as the UAV collects data. Zhang et al. [36], to explore the feasibility of a new generation of technology, embedded the YOLOv3 algorithm into the resource-limited NVIDIA Jason TX1 platform environment (see Figure 10) and had the UAV carry the embedded platform for real-time pedestrian detection experiments. The experimental results demonstrated the feasibility of implementing YOLO-based real-time target detection on a resource-limited mobile platform and provided a reference for the development of next-generation YBUT. Alam et al. [37], to alleviate the computational pressure on the onboard embedded processor of the UAVs and to enhance the practicality of YBUT, proposed a cost-effective aerial surveillance system that reserves the limited Tiny-YOLO computational requirements on the onboard embedded processor Movidius VPU, shifts the large Tiny-YOLO computational tasks to the cloud, and maintains minimal communication between the UAV and the cloud. Experimental results showed that the system is six-times faster in target detection processing at frames per second compared to the speed of other state-of-the-art approaches, while the application of airborne-embedded processor technology reduces end-to-end latency and network resource consumption (see Figure 11). Similar research was conducted by Dimithe et al. [38]. The

new generation of YBUT brings the YOLO algorithm and drone technology closer together. Although the new generation of YBUT does not show higher performance than previous technologies due to the limited computational resources of the embedded processor on board, the advantages of the new generation of YBUT were demonstrated with practical results by Zhang et al. [36]. It is sufficient to show that the future development of YBUT will tend towards a high degree of integration of YOLO algorithms with UAV technology.

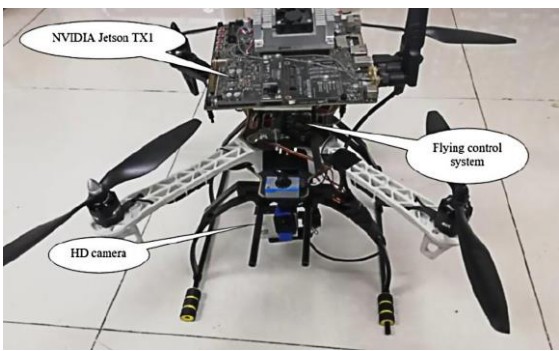

**Figure 10.** Four-rotor monitoring UAV [36].

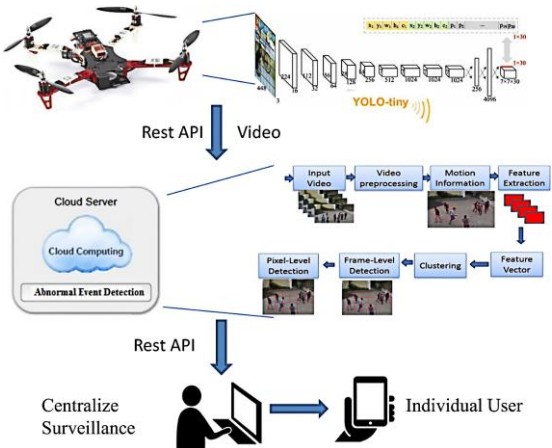

**Figure 11.** Complete system design by Alam et al. [37].

Based on this research, Cao et al. [39] proposed a target detection and tracking method based on the YOLO algorithm and the PID algorithm by using a new generation of high-performance embedded processor, NVIDIA Jason TX2, in combination with the pixhawk4 flight control processor. The PID algorithm performs UAV flight control, and the YOLO algorithm is used to identify objects, to extract pixel coordinates and then to convert the pixel coordinates to actual coordinates, where pixel coordinates were the coordinates of the target object relative to the camera image. The actual coordinates were the relative coordinates of the target in a spatial coordinate system constructed with the camera lens as the coordinate origin. Experimental results showed that the method can effectively detect flight targets and perform real-time tracking tasks. Doukhi et al. [40] used a UAV equipped with an Nvidia Jetson TX2 high-performance embedded processor and a PID controller. Then, they deployed the YOLOv3 algorithm in the embedded processor to intuitively guide the UAV to track the detected target by using the YOLO-based target detection algorithm, while the PID controller was used to control the UAV flight. Experimental results showed that the proposed method successfully achieves a visual SLAM for localization and UAV tracking flight through the fisheye camera only without external positioning sensors or the introduction of GPS signals (see Figure 12). Afifi et al. [41] proposed a robust framework for multiscene pedestrian detection, which uses YOLO-v3 object detection as the backbone detector (see Figure 13) and runs on the Nvidia Jetson TX2 embedded processor onboard

the UAV. Experimental results from multiple scenarios of outdoor pedestrian detection showed that the proposed detection framework showed better performance in terms of mAP and FPS, as the computational resources of the embedded processor increase compared to the YOLOv3 algorithm. To facilitate the development of a new generation of YBUT, Zhao et al. [42] improved YOLOv3-tiny, resulting in an 86.1% decrease in the model size, a 19.2% increase in AP50, and a 2.96-times faster detection speed than YOLOv3. The experimental results demonstrated that the improved algorithm is more suitable for low-end performance embedded processors in UAV target detection applications.

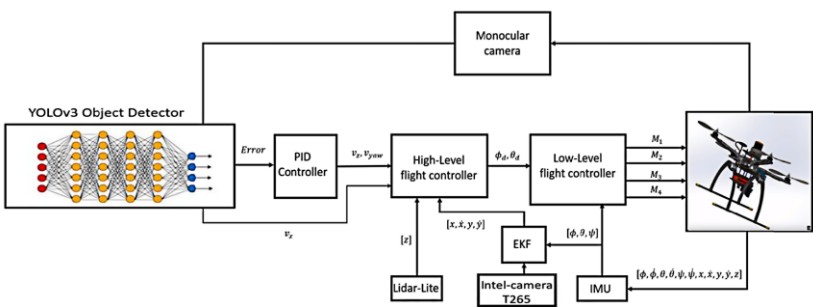

**Figure 12.** Software architecture for deep-learning-based motion control [40]. The red circles in the diagram represent the input RGB images in the YOLOv3 algorithm, the orange circles represent the calculation process of the YOLOv3 algorithm, and the blue circles represent the target and bounding box data detected by the YOLOv3 algorithm.

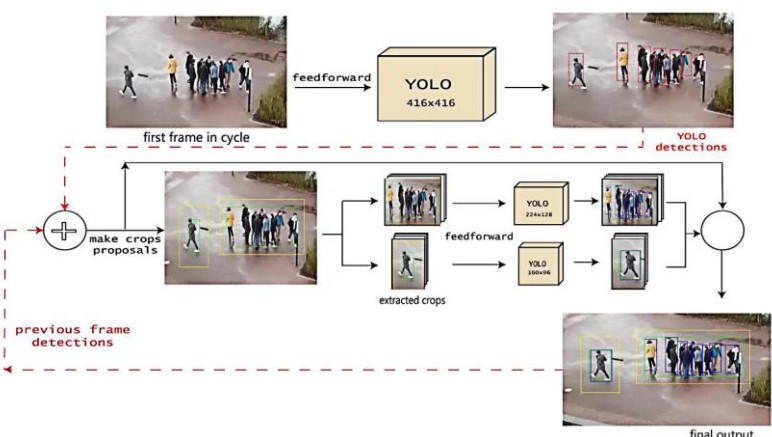

**Figure 13.** Workflow of the pedestrian detection framework [41].

Although the new generation of YBUTs can simplify the operational steps in applications, can improve operational integrity, and can increase the adaptability of technology applications, it is still difficult to obtain satisfactory performance for complex applications based on the limited computing power of existing embedded processors. At the same time, high-performance embedded processors have been slow in development and may not be able to match the computing performance of high-performance computer processors for some time. Therefore, most researchers prefer applications that take the form of a UAV collecting image or video data and a high-performance computer deploying YOLO for object detection [43]. Regardless of which of the two approaches researchers take, each study and application drives YBUT research forwards so that YBUT continues to be understood, accepted, and used by researchers in other fields.

## 4. Practical Applications of YBUT in Several Fields

In recent years, UAV load capacity, as one of the key points of UAV technology development, has achieved greater results, which provides the basis for carrying professional

equipment embedded with YOLO algorithms for object identification. The YOLO algorithm has been very widely used in various fields for object identification detection, while the method of carrying embedded processors in UAVs for object identification detection in the air has only just started to become popular. With the development of UAV technology and advances in algorithm performance, YBUT applications are expected to spread widely in life and in production in the coming years. Examples of YBUT applications in engineering, transportation, agriculture, automation, and other fields are outlined below, and these application methods or application approaches are discussed.

### 4.1. Related Applications in the Field of Engineering

Engineering is the main activity of everyday production and an important way of generating economic value. Amongst engineering operations, manual operations are both an important way of increasing productivity and a hindrance to it. Highly efficient large-scale manual operations in engineering are bound to produce higher production values, but there are also problems with overall production being affected by manual errors. With the advent of the industrial age, large-scale machine production has gradually replaced manual production, resulting in an exponential increase in output and a gradual reduction in costs. However, certain special jobs still need to adhere to manual work, such as checking power components of transmission lines and monitoring industrial instrumentation data. Although these jobs are not very difficult, the work is tedious, and it is very easy for staff to become fatigued and negligent, resulting in serious consequences. With the progress of UAV technology and the YOLO algorithm, some problems in engineering can be addressed by using a machine instead of manual labor or manual operation of the machine, which can alleviate the labor pressure on the staff to a considerable extent.

In engineering applications, YBUT has been successfully used and can, to a certain extent, replace people in some operations. The more mature research fields in which YBUT has been applied are transmission line detection [44], building surface detection [45], moving target tracking [46], gauge display reading [47], photovoltaic module detection [48], and building identification and classification [49]. According to the current survey, YBUT application research in the engineering field, researchers prefer the direction of transmission line detection. Objects, such as power line poles [50], insulators [51], electrical components [52], distribution line poles [53], transmission towers [54], bird nests [55], and breakers [56], can be accurately identified, classified, and located in complex environments. For example, Bao et al. [57] proposed an end-to-end parallel mixed attention detection YOLO network (PMA-YOLO) by collecting transmission line vibration damper data through UAVs and then creating a dataset to train and test the model; the results showed that the model can detect abnormal vibration dampers with an accuracy of 93.8% (see Figure 14). The successful detection and classification of various equipment and facilities in these transmission lines lay the technical basis for the construction of future intelligent power systems.

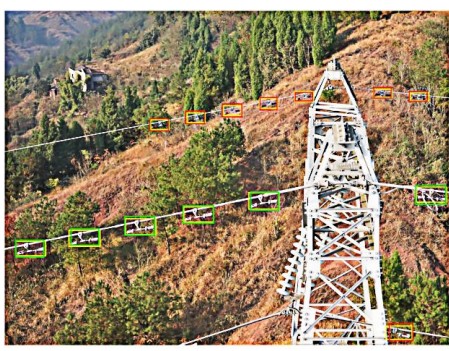 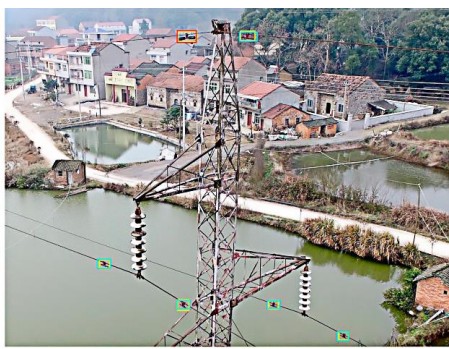 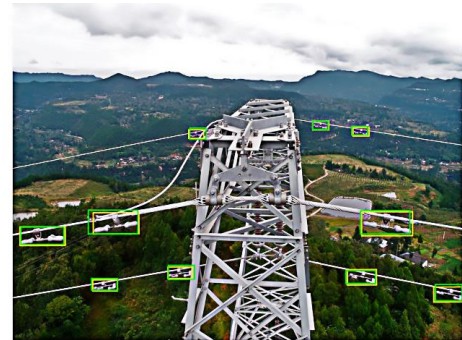

**Figure 14.** Experimental results of the PMA-YOLO network for the detection of anomalous vibration dampers [57]. The ground truth boxes and prediction boxes for "rusty", "defective", and "normal" dampers are shown in yellow, red, blue, and green, respectively.

Recently, Alsanad et al. [58] proposed an improved YOLOv3 algorithm for small UAV detection in low-altitude airspace; experiments showed that the disclosed improved model of the algorithm can effectively detect low-altitude UAVs in complex environments (see Figure 15) and can be successfully applied to the anti-drone research field to manage low-altitude airspace UAVs. The proposed method yielded a further enhancement in the low-altitude small-UAV detection performance of YBUT based on previous studies [59–62]. Other information regarding YBUT applications in the engineering field is shown in Table 2.

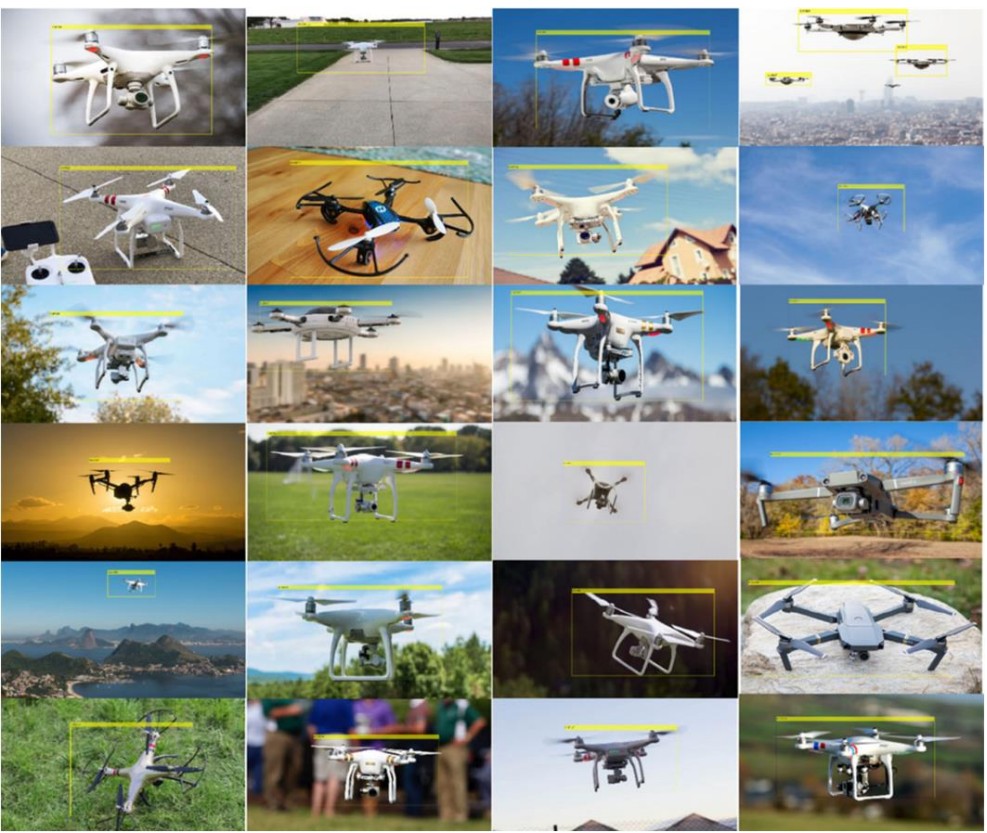

**Figure 15.** Results of the improved YOLOv3 algorithm for UAV detection [58].

**Table 2.** Overview of papers that explicitly address YBUT in engineering.

| YOLO Models | Reference | Object | Metric | Paper Type | Sensors | Purpose |
|---|---|---|---|---|---|---|
| YOLOv2 | Sousa et al. (2022) [63] | Ceramic detachment | Precision 99%, Recall 98% | Journal | Cameras | Building surface inspection |
| | Wang et al. (2019) [64] | Small objects | Accuracy 85% | Conference paper | Cameras | Algorithmic research |
| | Han et al. (2020) [65] | Insulators | Precision 92.1%, Recall 92.2% | Journal | Cameras | Transmission line inspection |
| | Yan et al. (2021) [66] | Electrical components | N/A | Conference paper | Cameras | Transmission line inspection |
| YOLOv3 | Liu et al. (2021) [67] | Insulators | Precision 94%, Recall 96% | Journal | Cameras | Transmission line inspection |
| | Kumar et al. (2021) [68] | Concrete damage | Accuracy 94.24% | Journal | Cameras | Building surface inspection |
| | Tu et al. (2021) [69] | Power towers and Insulators | Accuracy 88% | Journal | Cameras | Transmission line inspection |
| | Ding Lu et al. (2021) [70] | Holes and bolts | N/A | Journal | Cameras | Aerial manipulation platform |
| | Yang et al. (2022) [71] | Insulators | mAP 94% | Journal | Cameras | Transmission line inspection |
| YOLOv4 | Kim-Phuong et al. (2021) [72] | UAVs | Accuracy 87.37% | Conference paper | Cameras | Moving target tracking |
| YOLOv5 | Wang et al. (2021) [73] | Small objects | mAP 81.1% | Conference paper | Cameras | Algorithmic research |

YBUT is of great value to the engineering field and the productivity of society. Although there are still more engineering problems waiting to be resolved and more traditional manual methods waiting to be improved, YOLO algorithm object detection continues to become more accurate and faster; UAVs are becoming more convenient and safer, and if YBUT can continue to be used to develop innovations in the engineering field, then YBUT can create more value in the engineering field.

### 4.2. Related Applications in the Field of Transportation

With the expansion of human space and the extension of people's physical movement, the dependence on transport for daily travel is increasing. This has led to a dramatic increase in the size of roads and the number of vehicles over the last few decades. When there are many roads and many means of transport, their management becomes very important. The legislature has set up various traffic regulations to limit their use to ensure a stable order in life, but monitoring their compliance accurately and effectively is a problem that persists. Although there are cameras all over the streets and alleys, this does not fully detect all violations of the law and does not impose penalties.

To further manage and constrain the various modes of transportation in life, several attempts have been made in the field of transportation with YBUT. For example, Feng et al. [74] proposed a YOLOv3-based method for UAV detection (see Figure 16). Omar et al. [75] proposed an aerial image vehicle detection method based on the YOLOv4 algorithm (see Figure 17), and Liu et al. [76] proposed a method for the automatic detection and tracking of vehicles in an urban environment by UAVs based on the YOLOv4 and DeepSORT algorithms. These studies have yielded excellent results in motorized and non-motorized vehicle recognition and classification tasks based on datasets of air traffic images and have also enabled the automatic detection and tracking of urban vehicles. The accurate identification and classification of motor vehicles and non-motorized vehicles allows for accurate restraint of their behavior according to road management rules in intelligent traffic management, while the automatic detection and tracking of vehicles can provide assistance in the effective punishment of violations. The fundamental applied research on YBUT in urban traffic further accelerates the intelligent management of urban traffic and contributes to the creation of a civilized city.

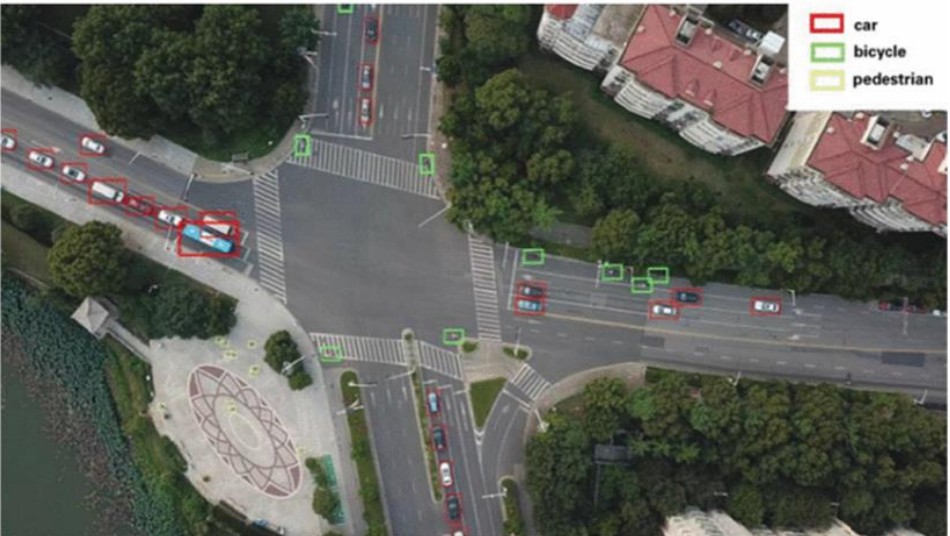

**Figure 16.** Vehicle detection results based on urban road videos [74].

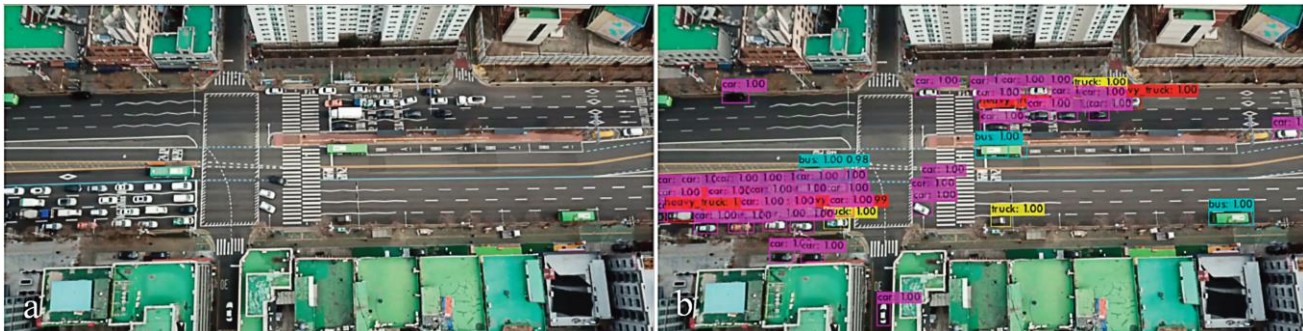

**Figure 17.** (**a**) UAV acquisition images, (**b**) UAV image detection results [75].

Both urban traffic management applications and urban road management are important directions for the application of YBUT technology. Silva et al. [77] designed a distributed UAV platform deploying YOLOv4 to detect road damage (see Figure 18). Zhao et al. [78] proposed a YOLOv3-based algorithm for UAV highway center mark detection, YOLO-Highway (see Figure 19). Recently, Ma et al. [79] proposed a new method for road damage detection based on YBUT, which experimentally showed better performance than previous similar studies and further promoted the application of road damage detection technology in urban road management. The intelligent management of urban traffic is not only the management of motor vehicles and non-motor vehicles but also the management of urban roads. The widespread application of YBUT in the field of traffic greatly promotes the process of intelligent management and has great significance for the convenience of future residents' lives. The expansion of the application of YBUT in urban road management is another step forward in the promotion of intelligent urban traffic management. Other information regarding YBUT applications in the transportation sector is shown in Table 3.

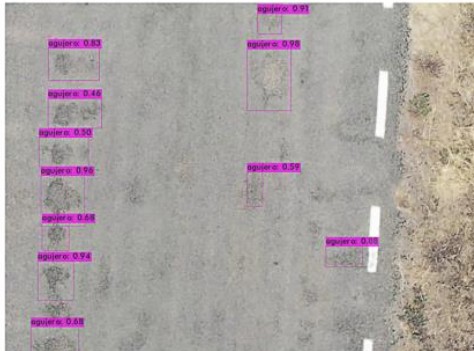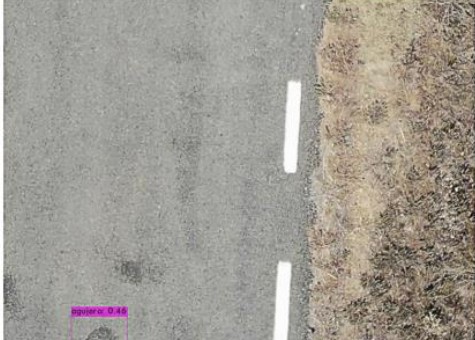

**Figure 18.** Road damage detection results [77].

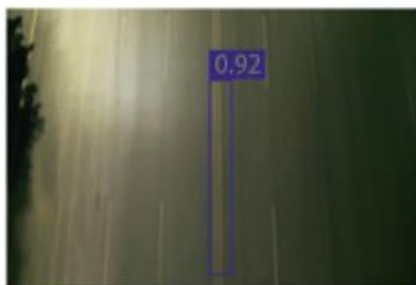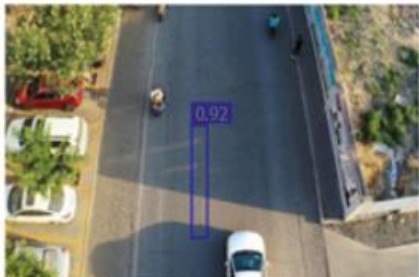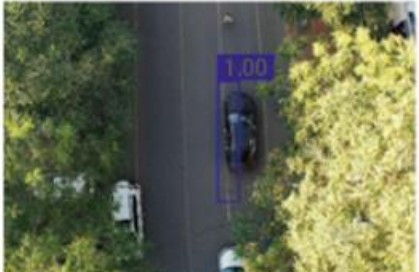

**Figure 19.** Detection results for road signs in various environmental conditions [78].

**Table 3.** Overview of papers that explicitly address YBUT in transportation.

| YOLO Models | Reference | Object | Metric | Paper Type | Sensors | Purpose |
|---|---|---|---|---|---|---|
| YOLOv2 | Kim et al. (2019) [80] | Road cracks | mAP 33% | Journal | Cameras | Road safety inspection |
| | Sharma et al. (2022) [81] | Railway tracks | Precision 74%, Accuracy 85%, mAP 70.7% | Journal | Cameras | Road safety inspection |
| YOLOv3 | Krump et al. (2019) [82] | Vehicles | mAP 64.4% | Conference paper | Cameras | Algorithmic research |
| | Luo et al. (2020) [83] | Vehicles | mAP 97.49% | Journal | Cameras | Algorithmic research |
| | Hassan et al. (2020) [84] | Road cracks | Accuracy 92%, mAP 90% | Conference paper | Cameras | Road safety inspection |
| | Chung et al. (2020) [85] | Vehicles | mAP 35.08% | Conference paper | Cameras | Algorithmic research |
| | Li et al. (2021) [86] | Vehicles | N/A | Journal | Cameras | Traffic Management |
| | Chen et al. (2021) [87] | Vehicles | mAP 50.05% | Journal | Cameras | Vehicle tracking and speed estimation |
| | Rampriya et al. (2022) [88] | Obstacles on the railway track | Precision 70.68%, Accuracy 70.83%, Recall 73.64% | Journal | Cameras | Road safety inspection |
| YOLOv4 | Gupta et al. (2022) [89] | Military vehicles | N/A | Journal | Cameras | Military vehicle detection and classification |
| | Golyak et al. (2020) [90] | Vehicles | N/A | Conference paper | Cameras, Thermal imager | Detection of unmanned vehicles |
| | Emiyah et al. (2021) [91] | Vehicles | N/A | Conference paper | Cameras | Vehicle detection and counting |
| | Luo et al. (2022) [92] | Vehicles | mAP 71.97% | Journal | Cameras | Algorithmic research |
| YOLOv5 | Feng and Yi (2022) [93] | Vehicles | mAP 89.74% | Journal | Cameras | Traffic Management |
| | Chen et al. (2022) [94] | Vehicles | Precision 91.9%, Recall 82.5%, mAP 89.6% | Journal | Cameras | Traffic Management |
| | Luo et al. (2022) [95] | Vehicles | mAP 85.35% | Journal | Cameras | Algorithmic research |

The above shows that researchers have made considerable research progress in this area and demonstrates the great potential of YOLO-based UAV application technology in the transport sector. With the support of this technology, not only can the cost of traffic video acquisition and processing be significantly reduced, but the spatial flexibility of traffic supervision is also enhanced. Although fewer researchers have experimented in this area, it is unlikely that the application of this technology is limited to scenarios, such as vehicle inspection and road detection; there must exist many more applications that are more beneficial to people's everyday lives.

*4.3. Related Applications in the Field of Agriculture*

In agriculture, there are often situations where failure to detect early symptoms of pests and diseases can lead to major pest and disease disasters and severe economic losses. When preventing or treating pests and diseases, there may also be excessive use of pesticides that can lead to environmental pollution and reduced crop yields. Wild vegetables, which are not commonly encountered every day, are often found in sites with lush vegetation, are less productive but have high nutritional value, and finding them has always been a serious challenge. We can use drones to perform some of the agricultural work and use the YOLO algorithm to assist in this process, which can be much more efficient and save time.

In this area of agriculture, many tricky jobs already have new solutions based on YBUT. With the continuous development and extension of YBUT, it is now possible to detect different targets and features among large plant species, such as in dead tree detection [96], pine wilt nematode disease detection [97–99] (see Figure 20), pine wilt detection [100], oil palm tree fruit detection [101], and other tasks. Additionally, YBUT can be applied in analyses involving small plants, such as in weed detection around peas and strawberries [102] (see Figure 21), field wheat phenotype monitoring [103], and tomato germinator detection [104]. Moving targets, such as animals, can also be detected, classified, and counted with high

accuracy [105] (see Figure 22). Other information on YBUT applications in the agricultural sector is given in Table 4.

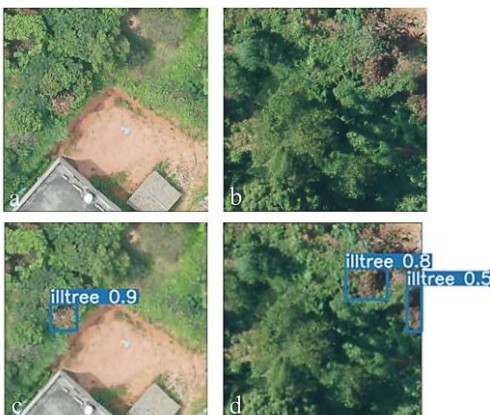

**Figure 20.** (**a**,**b**) are the original images of the diseased trees detection region, (**c**,**d**) are the results of the MobileNetv2-YOLOv4 algorithm for diseased trees detection of the region [97].

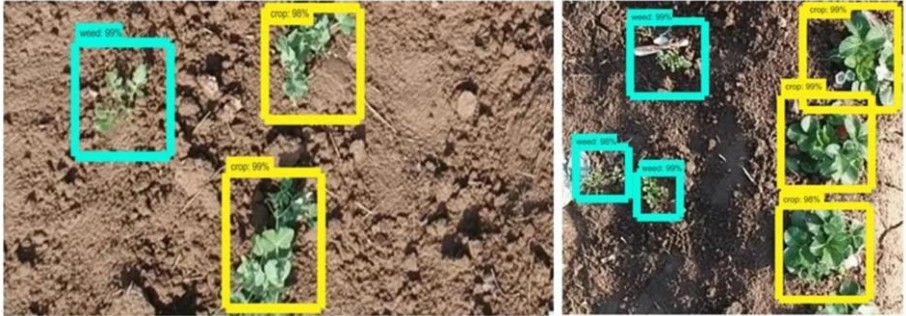

**Figure 21.** Weed identification results for pea crop area and strawberry crop area [102].

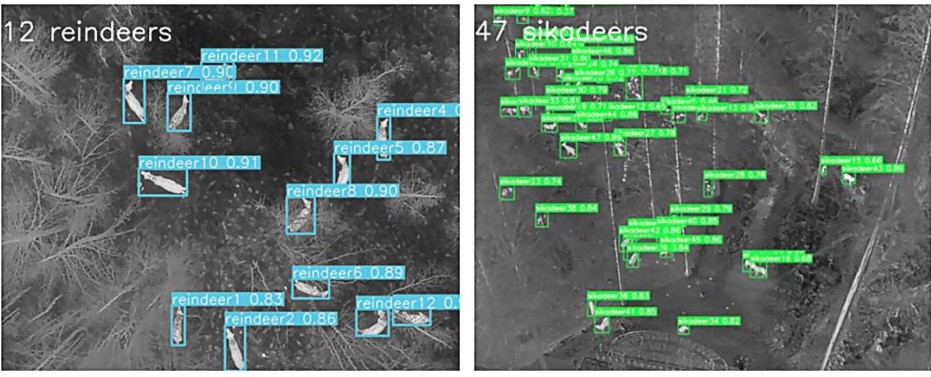

**Figure 22.** Detection counts of reindeer and sika deer by using the YOLOv5s improved model [105].

In this section, we provide an overview of the applications of YBUT in agriculture, and these exploratory applications point the way for expanding YOLO-based UAV applications in agriculture. Although the technology is still at the beginning stage in agriculture and many issues have not yet been resolved, such as dataset collection and sharing and stable YOLO algorithms that are more suitable for applications, we believe that YBUT can aid in the development of smart agriculture and has a broad scope of development in the agricultural field.

**Table 4.** Overview of papers that explicitly address YBUT in agriculture.

| YOLO Models | Reference | Object | Metric | Paper Type | Sensors | Purpose |
|---|---|---|---|---|---|---|
| YOLOv3 | Priya et al. (2021) [106] | Cattle | N/A | Conference paper | Cameras | Livestock management |
| | Ulhaq et al. (2021) [107] | Animals | mAP 87.1% | Journal | Thermal imager | Animal management |
| | Petso et al. (2021) [108] | Animals | F1 96% | Journal | Cameras | Wildlife monitoring |
| | Guzel et al. (2021) [109] | Wild mustard | Precision 45–99% | Journal | Cameras | Crop protection |
| | Hashim et al. (2021) [110] | Vegetation | Accuracy 84% | Journal | Multispectral camera | Hybrid Vegetation Detection |
| YOLOv5 | Idrissi et al. (2022) [111] | Burrow, Deadwood, Pine, Grass, Oak, Wood, Fire, Pedestrian | mAP 44.3% | Journal | Cameras | Evaluating the Forest Ecosystem |
| | Jemaa et al. (2022) [112] | Orchard tree | Precision 91% | Conference paper | Cameras | Orchard tree management |
| | dos Santos et al. (2022) [113] | Leaf-cutting ants | Accuracy 98.45% | Journal | Cameras | Optimizing the use of pesticides |
| | Puliti and Astrup (2022) [114] | Tree damage | Precision 76%, Recall 78% | Journal | Cameras | Evaluating the Forest Ecosystem |

### 4.4. Related Applications in the Field of Automation

The production method of the future is automated production with machines completely replacing manual labor. In everyday production, most operations require human control of the machines, while some of the more technologically advanced production operations have already been automated with machines replacing humans. In operations where staff are involved in production, their main task is to control the machine, i.e., to adjust the machine's working status according to the real-time operational situation. The combination of computer technology, which can now make decisions instead of humans, and object detection technology based on the YOLO algorithm, which can detect the status of the operation in real time and can provide feedback, can replace staff control of the machine to a certain extent. If both technologies are applied to drone platforms, it may be possible to reduce the labor pressure for workers and can increase the productivity of some industries.

To automate the use of YBUT in various applications, numerous researchers have developed different supporting technologies. After many studies, the technology for the detection, tracking, and avoidance of specific targets has matured and is now largely automated [115–118]. Notably, YBUT has been effectively used for the detection and localization of pedestrians [119–123]. Moreover, with an increase in unmanned mobility concepts, certain applications have been rapidly automated. Kraft et al. [124] proposed a YOLOv4-based method for locating litter in parks by using drones. The experimental results showed that the drones can detect litter and can collect litter location data in a fixed area while marking the litter location on a map for sweepers to see for easy cleaning (see Figure 23). In the future, the system can also work together with other equipment to locate and automatically sweep up litter, completely reducing the workload of sweepers. Liao et al. [125] proposed a UAV-based marine litter detection system that uses a UAV with an improved YOLO algorithm for marine litter detection; their system transmits the results to a ground-based monitoring platform via the internet to assist government agencies in implementing management plans (see Figure 24). Other information regarding YBUT applications in the automation sector is shown in Table 5.

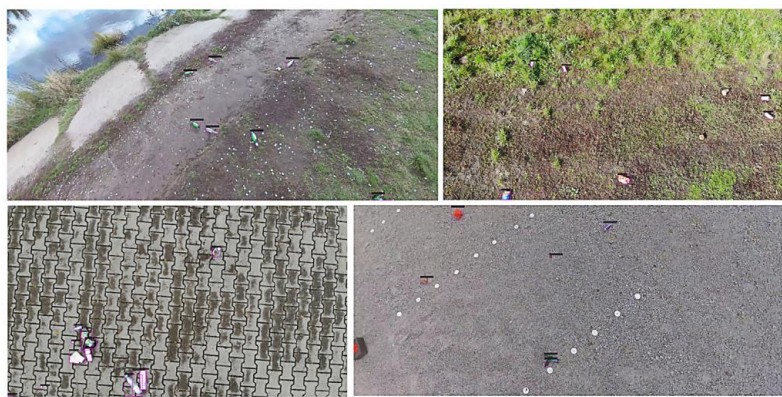

**Figure 23.** Results of UAV dataset detection using YOLOv4 [124].

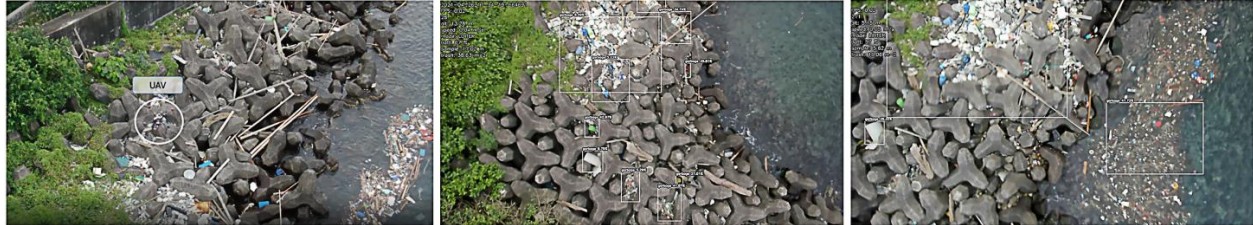

**Figure 24.** Results of UAV litter detection at Badouzi fishing port [125].

The successful application of YBUT in the field of automation demonstrates a viable method for achieving an unmanned and automated future. The research information obtained from existing exploratory practical applications provides a credible reference for reducing the pressure on human labor and increasing existing productivity levels in the future. In the coming period, the research focus of YBUT in the engineering field should be on expanding the types of operations to which the technology can be applied, developing specialist drones, developing high-performance YOLO algorithms suitable for embedded environments, and developing visual and convenient control systems.

**Table 5.** Overview of papers that explicitly address YBUT in automation.

| YOLO Models | Reference | Object | Metric | Paper Type | Sensors | Purpose |
|---|---|---|---|---|---|---|
| YOLOv3 | Liu et al. (2020) [126] | Small objects | mAP 72.54% | Journal | Cameras | Small object detection |
|  | Wang et al. (2020) [127] | UAVs | N/A | Journal | Cameras | Airport obstacle-free zone monitoring UAV system |
| YOLOv4 | Kong et al. (2022) [128] | Pedestrian | mAP 39.32% | Journal | Cameras | Pedestrian Detection and Counting |
| YOLOv5 | Maharjan et al. (2022) [129] | River Plastic | N/A | Journal | Cameras | Plastic waste management |

### 4.5. Related Applications in Other Fields

In addition to the main areas of YBUT application discussed above, some researchers have explored completely new areas, experimented with new methods, and used these methods to promote and enhance the applicability and usefulness of YBUT. Wyder et al. [130] integrated YBUT with vision service algorithms to successfully achieve the autonomous detection and tracking of moving targets in a GPS-limited environment. Quan, Herrmann et al. [131] proposed Project Vulture, an intelligent human–subject location system for UAVs based on the YOLO algorithm, and the system possessed higher sensitivity than other peer systems in mountain rescue operations. Similar content has been studied by Kashihara et al. [132]

and Sambolek and Ivasic-Kos [133]. Arnold et al. [134] investigated object classification functions and reactive group behavior in a dispersed autonomous heterogeneous swarm of UAVs deployed with YOLO; their approach supported the identification of specific targets with a UAV, and other UAVs were able to learn the behavior accordingly. The experimental results showed that the system still performs well at 25 m from the building. Jing et al. [135] proposed a neural network based on YOLOv5s-ViT-BiFPN, which can assess the damage of rural houses after natural disasters using drone images (see Figure 25). Information regarding YBUT applications in other areas is shown in Table 6.

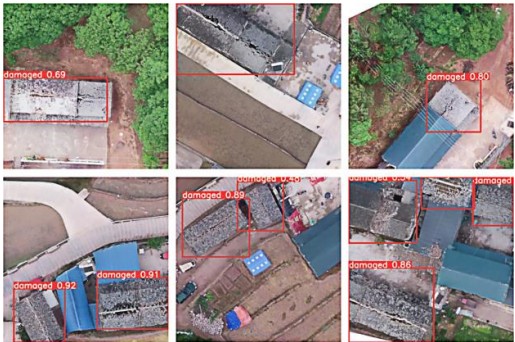

**Figure 25.** YOLOv5s-ViT-BiFPN algorithm for detecting damaged houses [135].

YBUT has been used successfully in various fields, which has greatly contributed to the development of the industry and the advancement of technology for the benefit of society. From the above overview of the various areas, there is much more value to be created by this technology. To create more value, we can improve and optimize existing applications, expand the idea with examples of successful applications, and use this to apply the technology to more areas.

**Table 6.** Overview of papers that explicitly address YBUT in other fields.

| YOLO Models | Reference | Object | Metric | Paper Type | Sensors | Purpose |
|---|---|---|---|---|---|---|
| YOLOv1 | Ajmera and Singh (2020) [136] | Missing victim | N/A | Conference paper | Cameras | Urban Search and Rescue |
| | Sudholz et al. (2022) [137] | Rusa deer | N/A | Journal | Cameras, Thermal imager | The detection and monitoring of invasive species |
| YOLOv2 | Opromolla et al. (2019) [138] | UAVs | N/A | Journal | Cameras | Visual-based detection and tracking of cooperative UAVs |
| | Merizalde and Morillo (2021) [139] | Pedestrian | Rcall 90% | Conference paper | Cameras | Real-time Social Distancing Detection |
| | Kim et al. (2019) [140] | Mobile construction resources | Accuracy 97.43% | Journal | Cameras | Protecting construction workers |
| | Hong et al. (2019) [141] | Birds | N/A | Journal | Cameras | Wildlife monitoring |
| | Arola and Akhloufi (2019) [142] | UAVs | N/A | Conference paper | Cameras | Collaborative UAV research |
| | Zheng et al. (2019) [143] | Distress personnel | N/A | Conference paper | Cameras | A Search and Rescue System for Maritime Personnel |
| | Silvirianti et al. (2019) [144] | UAV flight behavior | Accuracy 83% | Conference paper | Cameras | Search and rescue for people in distress in the forest |
| | Zhang et al. (2019) [145] | Sea surface ships | N/A | Conference paper | Cameras | Algorithmic research |
| | Medeiros et al. (2021) [146] | Human posture | N/A | Conference paper | Cameras | Human posture guidance system |
| YOLOv3 | Sarosa et al. (2020) [147] | Victims of natural disasters | Accuracy 89% | Conference paper | Cameras | Search and rescue for victims of natural disasters |
| | Rizk et al. (2021) [148] | Human | Accuracy 78.78% | Conference paper | Cameras | Search and rescue for victims of natural disasters |
| | Qi et al. (2021) [149] | Moving Target | N/A | Conference paper | Cameras | Moving target detection and tracking |
| | Panigrahi et al. (2021) [150] | Wildlife | mAP 95% | Conference paper | Cameras | Biodiversity analysis |
| | Wang et al. (2021) [151] | Offshore Small Targets | Precision 92.7%, Recall 92.06%, mAP 95.58% | Journal | Cameras | Algorithmic research |
| | Tanwar et al. (2021) [152] | Pedestrian | N/A | Journal | Cameras | Real-time Social Distancing Detection |
| YOLOv5 | Gromada et al. (2022) [153] | Military targets | N/A | Journal | Cameras, Synthetic aperture radar | Algorithmic research |
| | Bahhar et al. (2023) [154] | Wildfire and Smoke | mAP 85.8% | Journal | Cameras | Forest fire detection |

## 5. Development Prospects

With the rapid development of YOLO-based object detection technology and special UAV research, the YOLO-based UAV industry has set off a technological boom with multifield applications and multidirectional development. With the assistance of a variety of cutting-edge technologies, it is possible to improve productivity and quality of life and to create economic benefits while creating good ecological, environmental, and social benefits. YBUT shows increasingly obvious value and potential as it develops.

### 5.1. Improving the Quality of UAV Datasets and Training YOLO Algorithms Suitable for Aerial Imagery

In the practical application of YBUT, it is often the case that accuracy and speed are high for ground-based operational tests but low for aerial UAV operations, possibly due to the unsuitability of the dataset used by recent target detection algorithms [155–158]. Therefore, when collecting and selecting datasets to create models that perform as well as possible for the UAV operational environment, the following should be noted: (1) When performing image acquisition of the target, care should be taken that the acquisition equipment is as consistent as possible and that the same equipment is used for acquisition from start to finish so that the same-resolution image can be obtained. This helps ensure that the image content is not distorted due to inconsistencies in image size during algorithm training. (2) The dataset should be collected from as many different angles as possible, e.g., different camera angles, weather differences, various light intensities, several poses of the target, numerous target positions, and different target backgrounds. (3) When annotating the target border category and coordinate information within the dataset, we need to reduce the area of the background content within the border as much as possible and must ensure that all the target content is placed within the border. When annotating multiple categories of borders, we need to minimize the area of overlapping borders to avoid the algorithm combining the two into the same content.

At the same time, to accelerate the progress of research on YOLO-based UAV object recognition technology, it is recommended that most developers create good-performing models, describe the content and application performance of the datasets used, and upload them to the community for sharing.

### 5.2. Research into Object Detection Algorithms Suitable for UAV-Embedded Processors

In YBUT research, UAVs can carry limited hardware resources for mobile processors and cannot be better transplanted to existing YOLO algorithms for application. Therefore, lightweight target detection algorithms should be investigated for mobile processors with limited resources, or performance optimization or network pruning model improvements should be made to existing YOLO algorithms [159–164]. Liu et al. [165] proposed a Slice-Concat structure based on YOLOv3 and YOLOv3-SPP, which can improve the target detection speed by simply changing the width and height of the uniform input dataset. Zhang et al. [166] proposed an intelligent approach for UAVs that combines machine learning, traditional algorithms, and intelligent AI algorithms. The YOLOv3 algorithm is then used to sense the location of objects in the environment and to classify them, and finally, AI is used to evaluate the working state. Experiments showed that the method has high computational speed and recognition accuracy, good generality, portability, and scalability, and they proposed a new development direction for future UAV technology.

To quickly promote the YBUT and facilitate learning and application by other interested researchers or industry beginners, it is recommended that all peer researchers who have successfully implemented the application for lightweight UAV models disclose their optimization methods and model source code and provide detailed explanations of the optimized parts and the optimized network structure.

### 5.3. Developing Professional, Stable, and Reliable UAVs in Combination with the YOLO Inspection Environment

Given the many areas of development of YBUT and the issues with UAVs themselves, further development and promotion of UAVs with high professionalism, high environmental adaptability, stability, and reliability should be carried out [167–172]. The overall design of UAVs should be combined with the field of UAV operations, while considering the environment, to improve the professionalism of UAV operations and to ensure the adaptability and stability of UAV operations. In terms of the UAV power system, we should develop the core components of power motors or engines, improve the service life of the core components, and reduce the total amount of the whole aircraft to improve the practicality of UAVs. In terms of UAV onboard equipment, multisensor fusion technology should continue to be developed and applied autonomously. In terms of UAV safety, the development of UAV safety flight algorithms is necessary. We also improve the UAV runaway self-protection system, realize effective obstacle avoidance, runaway self-protection, fault self-testing, and runaway warning functions. Moreover, we need to monitor all parameters of the UAV itself to protect the users' property.

### 5.4. Enhancing the Security of YBUT for Multiple Application Scenarios

The development of YBUT and its widespread use in various fields have led to the technology being gradually recognized by researchers, but in the pursuit of rapid technological development, safety issues are often easily overlooked. In the daily application of YBUT, UAVs mainly transmit data with ground control terminals by wireless communication, which is easily interfered with and invaded by others, thus causing problems, such as loss of control of UAVs and data information leakage [173]. In many application scenarios, if a UAV is hacked by others and loses control of its flight, it will not only pose a threat to the UAV itself but also to the surrounding environment and may even endanger the personal safety of the operator. To prevent relevant security issues from occurring, it is vital to enhance the data security of the YBUT. Both the storage and transmission of data information and the transmission of UAV movement control commands should be the main targets for security enhancements in the YBUT.

### 5.5. Popularising YBUT Knowledge, Training Technical Application Talents, and Improving Relevant Laws, Regulations, and Codes of Practice

Popularizing the knowledge of UAVs and YOLO algorithms and training composite talents in UAV control and YOLO algorithm application should be the role of higher education institutions, research units, relevant enterprises, and group organizations. Improving relevant laws, regulations, and codes of practice is an inescapable responsibility of the relevant legislative bodies in the face of the rise of new technological developments and applications. Although YBUT is developing rapidly, it takes time to achieve autonomous unmanned operation of drones, so talent for drone control should be cultivated to improve the overall level of the industry and the scope of production and use in life. At the same time, to adapt to the rapid iteration of YOLO algorithm versions, knowledge of YOLO algorithm applications should be popularized, and the ability of relevant personnel to apply YOLO algorithms should be improved to bring into play the diversity of YOLO-based UAV operations. To further protect the legal rights of users and others, the YBUT must be applied in strict accordance with relevant laws and regulations, and operators must be trained to ensure the correct use of the YBUT.

## 6. Conclusions

In any period, social progress needs advanced productivity as a basis, and every advancement needs time to develop. When an emerging field becomes popular, the field then gathers most of the current resources to develop it so that it rapidly progresses and spreads to other fields. Then, having been fully integrated with other fields, it is

presented to people in the way of practical applications to address the needs of life so that people benefit.

In this literature review, we demonstrate that the combination of deep learning YOLO algorithms and UAV technology can be of great use in the future, and an attempt is made to introduce and to promote the technology to attract the attention of more researchers. In this paper, we first describe the development of YBUT, including the early developments and leaps and bounds in the application of YOLO algorithms in conjunction with UAV technology. Second, to promote YBUT, the main areas where researchers have applied the technology and the recent state of research, as well as the exploration and experimentation of the technology in certain new areas, are presented. It is clear from the article that YOLO-based object detection algorithms could be a key enabler for future drone applications, allowing drones to provide better productivity and greater convenience.

Currently, UAV technology and YOLO-based object detection are relatively well established in their respective pre-existing fields, and the cross-fertilization of the two into new technologies is becoming an increasingly important area. The results show that there is a high degree of advantageous complementarity between UAV-derived aerial platforms and YOLO algorithms for object detection. However, the application methodology and performance of YBUT need to be further enhanced. The development of YBUT has, thus far, seen more applications in engineering, transportation, agriculture, and automation and less practice in other fields; the diffusion of the technology remains a challenge. The future development of technology needs to take these four issues into account. The actual detection performance of ground-acquired datasets applied directly to the training of UAV-based object detection algorithms is not very satisfactory, and further dedicated high-quality datasets need to be acquired. Deploying existing YOLO algorithms directly to mobile processors through optimization can complete the current exploratory research goals, but this step is still a long way from industrializing YBUT and requires dedicated algorithms to be developed for the UAV hardware environment. For future applications of the technology in more areas, a single specialist drone should be developed for specific use environments. The timely development of talent for the development and application of YBUT is also an effective way to rapidly promote the technology. To some extent, the rapid diffusion of YBUT and the continuous identification of new problems and needs during the diffusion process, addressing new problems and meeting new needs, can also contribute to technological progress.

**Author Contributions:** Conceptualization, C.C. and Z.Z.; methodology, C.C.; software, S.F.; validation, C.C., Z.Z. and S.G.; formal analysis, T.X.; investigation, C.C.; resources, C.C.; data curation, Z.Z.; writing—original draft preparation, C.C. and Z.Z.; writing—review and editing, W.Y.; visualization, C.C. and Z.Z.; supervision, Y.L.; project administration, T.X. and W.Y.; funding acquisition, T.X., W.Y. and Y.L. All authors have read and agreed to the published version of the manuscript.

**Funding:** This research was supported by the General Program of Liaoning Provincial Educational Department (LJKMZ20221059), the Key Tackling Program of Liaoning Provincial Educational Department (LSNZD202005), and the 111 Project (D18019).

**Data Availability Statement:** Not applicable.

**Acknowledgments:** We sincerely thank Dongxu Su, Yihan Liu, Hongyang Zhou, and Ziqi Yu from Shenyang Agricultural University for supporting this work.

**Conflicts of Interest:** The authors declare no conflict of interest.

## Nomenclature

| Acronyms | Definition |
| --- | --- |
| UAV | Unmanned aerial vehicle |
| YOLO | You Only Look Once |
| YBUT | YOLO-based UAV technology |
| UAVs | Unmanned aerial vehicles |
| FPS | Frames Per Second |
| mAP | Mean average precision |
| Faster R-CNN | Faster regional Convolutional Neural Network |
| RFCN | Region-based Fully Convolutional Network |
| SNIPER | Scale Normalization for Image Pyramids with Efficient Resampling |
| IoU | Intersection over union |
| F1 | Harmonic mean of precision and recall |
| AP | Average precision |
| VPU | Vision Processing Unit |
| PID | Proportional Integral Differential |
| SLAM | Simultaneous localization and mapping |
| GPS | Global Positioning System |
| DEEPSORT | Deep Simple Online and Realtime Tracking |

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
