# Peer review of "YOLO-Based UAV Technology: A Review of the Research and Its Applications"

_drones, doi:10.3390/drones7030190_

Round 1
Reviewer 1 Report
This paper presents an overview of YOLO-based UAV technology, referencing relevant research. However, there are some areas where the paper could be improved and clarified.
Firstly, the paper should provide more information about the meaning of "working mode" in the introduction. Additionally, the paper should compare YOLO with other object detection methods and explain why YOLO was chosen for this research. The paper provides a brief overview of YOLO's process (uniform resizing and cell division) but should provide more detail.
The paper's chapter 3 requires more detailed explanations of the presented methods. For instance, Figure 8 requires further explanation, and this part should clearly explain how YOLOv2 differs from the previous version. On page 10, it should be clarified what the difference between pixel coordinates and actual coordinates is in Cao et al.'s research. The paper should also provide more details on what was compared in Afifi et al.'s research when it mentioned that YOLO had better performance.
Overall, the paper needs more detailed explanations of the methods and terms used.
Author Response
Thank you for your letter and for the reviewers’ comments concerning our manuscript. We have studied comments carefully and have made correction which we hope meet with approval. The detailed responses have been uploaded via a document.

Reviewer 2 Report
Please read the attachment. Thank you.

Author Response

(The authors gave the same response as above.)

Round 2
Reviewer 1 Report
Dear authors,
I appreciated the improvement done on this revision. The clarity of the paper is improved. The manuscript can be published as it is.